# Crystal Structure, Raman Spectroscopy and Optical Property Study of Mg-Doped SnO$_2$ Compounds for Optoelectronic Devices

K. K. Singha [1], P. P. Singh [2,*] , R. Narzary [1], A. Mondal [3] , M. Gupta [4], V. G. Sathe [4], D. Kumar [4] and S. K. Srivastava [1,*]

[1] Department of Physics, Central Institute of Technology Kokrajhar, Kokrajhar 783370, India
[2] Department of Computer Science and Engineering, Central Institute of Technology Kokrajhar, Kokrajhar 783370, India
[3] Department of Chemistry, Central Institute of Technology Kokrajhar, Kokrajhar 783370, India
[4] UGC-DAE Consortium for Scientific Research, University Campus, Khandwa Road, Indore 452001, India
[*] Correspondence: pankajp.singh@cit.ac.in (P.P.S.); sk.srivastava@cit.ac.in (S.K.S.)

**Abstract:** Researchers have been consistently looking for new materials that can be integrated in optoelectronic and spintronic devices. In this research, we investigated the crystalline structure, Raman, and optical characteristics of Mg-doped SnO$_2$ compounds. The solid-state reaction technique was utilized to produce polycrystalline samples of Sn$_{1-x}$Mg$_x$O$_2$ ($0 \leq x \leq 0.10$) for their potential use in optoelectronics devices. It was discovered that all the compounds were synthesized into a tetragonal rutile-type structure of SnO$_2$. The analysis of these samples using Raman spectroscopy provided more evidence, supporting the creation of the tetragonal rutile phase of SnO$_2$ and the successful integration of Mg ions in SnO$_2$. The measurements of the optical properties, such as absorbance and transmittance, carried out with a UV-Vis spectrophotometer demonstrated that the optical band gap widened with the increase in the magnesium doping concentration in SnO$_2$. In addition, it was noticed that increasing the quantity of magnesium doping concentration led to an increase in the transmittance value from 83% to 91%.

**Keywords:** tin oxide; Mg doping; crystal structure; Raman spectroscopy; optical band gap

## 1. Introduction

During the last couple of decades, a great deal of research has been undertaken to develop a new family of novel materials that can be incorporated into optoelectronic and spintronic devices. The interest in such devices lies due to fact that they are thought to exhibit better performances than traditional semiconductors in several ways, such as being non-volatile, analyzing data in a faster way, consuming less power, and having more storage space [1–3]. Optoelectronic devices are those that make use of the features of both optics and electronics in their operation. Optoelectronics mainly focuses on how light reacts with electronic materials, especially semiconductors. Optoelectronic devices are an essential part of information technology and one of the most exciting research areas in modern optoelectronics and microelectronics. Transition metal (TM)-doped semiconducting oxides, commonly referred to as diluted magnetic semiconductor oxide (DMSO), were one of the materials subjected to significant research for use in spintronics and optoelectronics applications. DMSOs reveal ferromagnetic [4–8], optoelectronic [9–12], and many other features [13]. It was challenging to determine the origin of magnetism in these materials due to the development of a secondary phase of transition metal ion clusters. Researchers could not determine whether the room-temperature ferromagnetism (RTFM) in DMSO materials was due to the presence of magnetic ion clusters or was inherent to the material itself [14]. Therefore, research was done on several other materials to create a homogeneous, pure material displaying RTFM without any transition metal atoms. Recently, researchers have put their effort into developing non-magnetic-element-doped semiconducting oxide

materials with appropriate doping concentrations that exhibit various interesting properties, such as optical, magnetic, and transport properties. In addition, numerous un-doped oxides, including $ZrO_2$, $HfO_2$, $Al_2O_3$, $TiO_2$, and $In_2O_3$, were also found to exhibit ferromagnetism, which is typically caused by defects [15–19]. Bouzerar et al. [20] suggested a theoretical model where non-magnetic elements of group 1A of the periodic table and other elements can be substituted in the oxide matrix to customize $d^0$ ferromagnetism. Various ab initio studies have predicted $d^0$ ferromagnetism in non-magnetic-material-doped oxides with high Curie temperatures ($\theta_C$), namely $ZrO_2$, $SnO_2$, $TiO_2$, $ZnO$, $MgO$, $HfO_2$, and $In_2O_3$. Non-magnetic elements doped and co-doped semiconducting oxides materials, such as $SnO_2$, $TiO_2$, $HfO_2$, $CaO$, $ZnO$, and $ZrO_2$ [20–56], have been investigated, and a variety of fascinating features, including optical, magnetic, and transport properties have been identified.

Tin oxide, also known as stannic oxide ($SnO_2$), is an oxide semiconductor with a wide optical band gap (3.6 eV), greater transparency (less scattering), excellent electrical conductivity (less resistance), and more excellent chemical stability. These characteristics make the material suitable for gas sensors, flat-panel displays, catalysts, light-emitting diodes, solar cells, and other optoelectronic devices [48]. Crystals of tin oxide have a rutile structure (space group $P4_2/mnm$), which is a characteristic of the element. Each atom of tin can be broken down into coordination with six atoms of oxygen, and each atom of oxygen can be broken down into coordination with three atoms of tin. It demonstrates n-type conductivity with a broad direct band gap measuring 3.6 eV [49]. Many studies have looked at the crystal structure and magnetic and optical properties of $SnO_2$ doped with either a monovalent or divalent non-magnetic element, which reveal some fascinating facts. For example, spray-pyrolyzed transparent conducting films of Li-doped $SnO_2$ were studied by M. Mehdhi et al. [50] for their structural, optical, and electrical properties. It was discovered that for high-acceptor-doped films, the value of $E_g$ shifts towards lower energy (longer wavelengths), falling within a range of 4.1 to 3.61 eV. Wang et al. [51] also examined the $d^0$ ferromagnetism and band gap expansion in Li-doped epitaxial $SnO_2$ films. When looking at the films' ferromagnetic properties, they found that the thin film of 12% Li-doped $SnO_2$ with the most significant band gap had the highest saturation magnetization, measuring at 7.9 emu/cm³. Srivastava et al. studied Li-doped $SnO_2$ compounds. It was observed that 6% and 9% Li-doped compounds were found to exhibit low-temperature ferromagnetic ordering [25]. In another study on K-doped $SnO_2$ samples, it was noticed that a specific doping concentration led to a magnetic moment of the order of 0.2 $\mu_B$/K/ion at 2.5 K [52]. Mazumder et al. studied the photoluminescence and absorption features of high-Mg-doped $SnO_2$. It was noticed that the incorporation of Mg in $SnO_2$ widened the band gap ($E_g$) consistently [54]. Wu et al. reported the magnetic properties of Mg-doped $SnO_2$ epitaxial thin films, and they observed room-temperature ferromagnetism for 6% Mg-doped $SnO_2$ samples. The optical band gap of these materials was found to increase with Mg doping. The observed ferromagnetism was claimed to be induced by the holes created by Mg on the substitutional site [22]. He et al. [55] studied the effect of Mg doping on the structural, optical, and electrical properties of $SnO_2$ thin films deposited using the electron beam technique. It was observed that these films exhibited optical transmittance of 83% in the visible region. The band gap of the films was found to change from 3.49 to 3.78 eV [55]. Ali et al. [56] studied the effect of Mg doping on the structural and optical properties of aerosol-assisted chemical-vapor-deposited $SnO_2$ thin films. It was noticed that the optical transmission increased from 54% to 78%. The optical band gap was found to decrease for the initial doping concentration, followed by an increase on further increasing the Mg doping. Narzary et al. reported that $Sn_{0.94-y}Ag_{0.06}Mg_yO_2$ (with y = 0, 0.03, 0.06, 0.09, 0.12) exhibited RTFM with coercivity values lying within the range of 10–50 Oe [27]. These materials were found to show excellent optical transparency [27]. In the present work, we decided to conduct an experimental investigation into the crystal structure, Raman spectroscopy, and optical properties of $Sn_{1-x}Mg_xO_2$ (with x = 0, 0.02, 0.04, 0.06, 0.08, and

0.10) compounds. The current study involved the synthesis of the materials in bulk form via a solid-state route method to reduce fabrication faults and any characterization imprecision.

## 2. Materials and Methods

### 2.1. Preparation of the Materials

Solid-state reaction synthesis was used to prepare a polycrystalline compound of $Sn_{1-x}Mg_xO_2$ (with x = 0, 0.02, 0.04, 0.06, 0.08, and 0.10) from high-purity compounds of $SnO_2$ (99.9% purity supplied by Thermo Scientific, Waltham, MA, USA) and MgO (99.95% purity supplied by Alfa Aesar, Tewksbury, MA, USA) as the starting compounds. Using a digital balance with the model number BSA423S-CW manufactured by Sartorius (Goettingen, Germany), the appropriate quantity of each high-purity starting chemical compound ($SnO_2$ and MgO) was weighed using the previously determined stoichiometric ratio. After that, the combination that included the necessary compounds was thoroughly ground using an agate mortar and pestle, and acetone was added to the mixture to be more homogenous. After the starting materials were ground, the samples were annealed in powder form at four different temperatures: 200 °C for 10 h, 400 °C for 10 h, 500 °C for 20 h, and 800 °C for 20 h. Since MgO has a melting point of 650 degrees Celsius, pre-annealing was conducted to prevent any melting of Mg. In the final step, the newly created compounds were annealed in pellet form for five hours at 1000 degrees Celsius in the air.

### 2.2. Characterization Methods of the Materials

(i)      Structural Characterization

An XRD machine (D8 Advance XRD) was used in the scan range of 20°–80° with radiation from a $CuK_\alpha$ source so that the crystal structure and phase purity of the produced samples could be investigated. When operating with $CuK_\alpha$ radiation having an average wavelength of 1.5418 Å, the X-ray tube in the diffractometer had its parameters set to a 40 kV voltage and a 40 mA current. For the scan range of 2θ, a step size of 0.02° was selected as the optimal option. The surface morphological study was performed using scanning electron microscopy (FE-SEM, Sigma, Zeiss, NY, USA). Raman spectra were collected at 300 K using a diode laser with a wavelength of 473 nm using a micro-Raman spectrometer (Horiba-Jobin-Yvon, Oberursel, Germany, HR800 LABRAM) in the spatial resolution of $-257$ μm × 71 μm. The spectral resolution for the Raman measurements was $0.68$ cm$^{-1}$. The acquisition time for every spectrum was 12 s. The spectrum was collected without any baseline correction.

(ii)      Optical Properties Characterization

The samples were analyzed with a Shimadzu, Kyoto, Japan, UV-2600 UV-vis spectrophotometer, which was used to evaluate the transmittance and absorbance spectra of the optical medium for wavelengths ranging from 300 to 800 nm. Barium sulphate ($BaSO_4$) was mixed with the powder samples, and each sample was placed in the UV-Vis spectrophotometer apparatus using the solid-state probe mode. Each sample's absorbance spectra were obtained after adjusting for the $BaSO_4$ reference (base) line.

## 3. Results and Discussion

### 3.1. Crystal Structure

Figure 1 displays the XRD patterns of the synthesized compounds, i.e., $Sn_{1-x}Mg_xO_2$ (with x = 0, 0.02, 0.04, 0.06, 0.08, and 0.10), measured at room temperature (30 °C). According to the information on JCPDS Card No. 14-1445, both undoped $SnO_2$ and Mg-doped $SnO_2$ compounds exhibit a well-tetragonal rutile-type structure associated with the $P4_2/mnm$ space group of $SnO_2$. Within the range of the XRD, there was no detection of any additional diffraction peaks, which indicates that the samples are formed in a single phase, and no secondary phase is present, i.e., it does not include any crystalline parasitic phases. Moreover, it indicates that the samples are solvable up to a 10% Mg doping concentration.

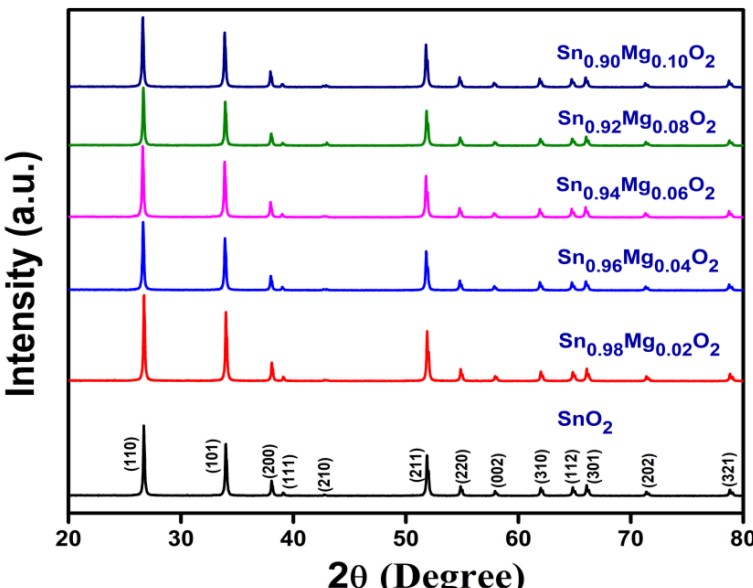

**Figure 1.** X-ray diffraction (XRD) patterns of $Sn_{1-x}Mg_xO_2$ (with x = 0, 0.02, 0.04, 0.06, 0.08, and 0.10) compounds.

The XRD patterns of the $Sn_{1-x}Mg_xO_2$ compounds (with x = 0, 0.02, 0.04, 0.06, 0.08, and 0.10), as refined via the Rietveld refinement method [57] using the Fullprof programming software-2021 [58], are displayed in Figure 2. Scale factors, background parameters, instrumental zero-point, lattice parameters (a, b, and c), peak shape parameters, individual anisotropic thermal and occupancy of atoms, and many more aspects were taken into account and refined in a systematic step-by-step approach during the Rietveld refinement method (using Fullprof software-2021), ensuring that the output accurately reflected a reliable value. In this illustration, the experimental data are depicted as circles, while the computed intensities are depicted as solid lines. The lines at the bottom of the graph show the difference between the observed and estimated intensities. This diagram denotes the permitted Braggs positions for the $P4_2/mnm$ space group as short vertical lines. It can be observed that the Rietveld software's computed XRD data and the experimental XRD data line up exactly with one another.

The calculated lattice parameters' values for the undoped $SnO_2$, a = b = 4.7277 Å and c = 3.1797 Å, are very similar to those reported in another study [52]. With an increasing Mg doping concentration, the lattice parameters and the cell volume estimated by refining XRD patterns with the Fullprof program indicate that the lattice parameters and cell volume initially increase for 2% Mg doping, followed by a decrease in the value. However, a scrutiny of the variation indicates that the changes are minimal and only at the third decimal place, as demonstrated in Figure 3 and listed in Table 1. It can be understood in terms of the ionic radii of $Sn^{4+}$ (0.69 Å) and $Mg^{2+}$ (0.72 Å), which are almost identical, and hence, the lattice parameters remain unaffected. The nominal decrease and irregular trend of variation in the lattice parameters might have arisen due to different amounts of cationic or oxygen vacancies. Table 1 provides the values of the lattice parameters, unit cell volume, and reliability factors ($R_p$, $R_{\omega p}$, $R_{exp}$, $R_{Bragg}$, $R_f$, and $\chi^2$) that were derived from the Rietveld analysis of XRD patterns. The mean crystallite sizes ($S_c$) of all the produced polycrystalline compounds were calculated by using Debye–Scherrer's formula:

$$S_c = \frac{k\,\lambda}{\beta\,cos\theta} \tag{1}$$

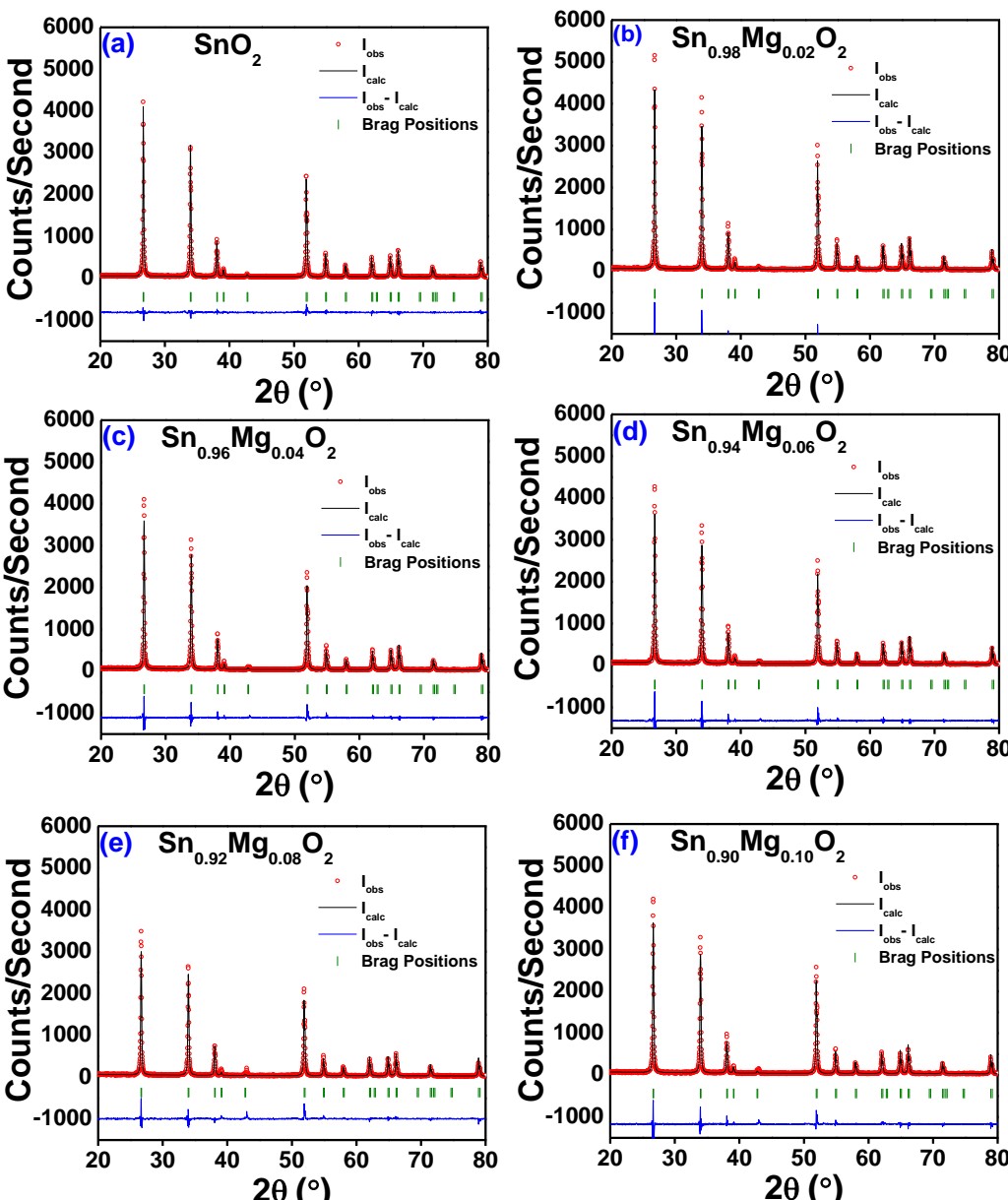

**Figure 2.** Rietveld refinement XRD patterns of the $Sn_{1-x}Mg_xO_2$ compound for the values of x equal to 0, 0.02, 0.04, 0.06, 0.08 and 0.10 and presented as (**a**), (**b**), (**c**), (**d**), (**e**), and (**f**) respectively.

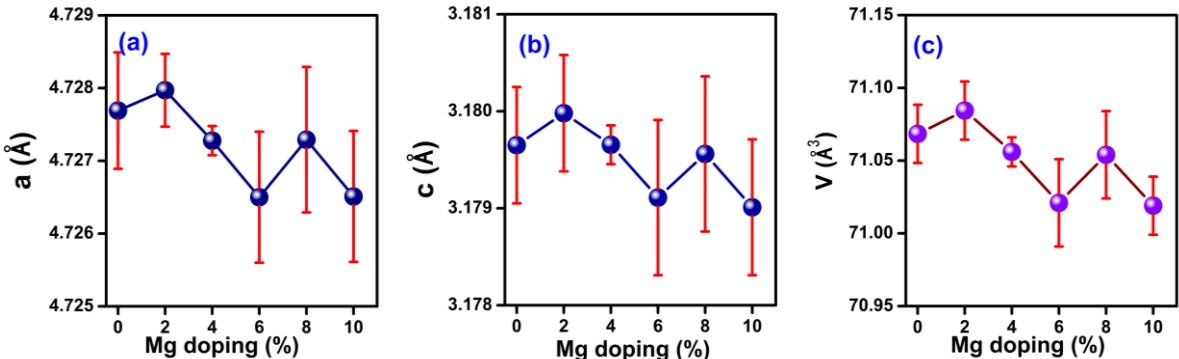

**Figure 3.** The variation in the lattice parameters (**a**,**b**) and the cell volume (**c**), together with the standard error bar, for $Sn_{1-x}Mg_xO_2$ (x = 0, 0.02, 0.04, 0.06, 0.08, and 0.10) compounds.

**Table 1.** Crystallographic parameters of $Sn_{1-x}Mg_xO_2$ (x = 0, 0.02, 0.04, 0.06, 0.08, and 0.10) compounds. The standard error for both the lattice parameters and the unit cell volume are shown in brackets. The $R_p$, $R_{\omega p}$, $R_{exp}$, $R_{Bragg}$, $R_f$, and $\chi^2$ indicate the reliability factors. $S_C$ represents the crystallite size.

| Sample/ Parameters | x = 0.00 | x = 0.02 | x = 0.04 | x = 0.06 | x = 0.08 | x = 0.10 |
|---|---|---|---|---|---|---|
| Space Group | Tetragonal ($P4_2/mnm$) | Tetragonal ($P4_2/mnm$) | Tetragonal ($P4_2/mnm$) | Tetragonal ($P4_2/mnm$) | Tetragonal ($P4_2/mnm$) | Tetragonal ($P4_2/mnm$) |
| a (Å) | 4.7277 (0.0008) | 4.7280 (0.0005) | 4.7273 (0.0002) | 4.7265 (0.0009) | 4.7273 (0.0010) | 4.7265 (0.0009) |
| b (Å) | 4.7277 (0.0008) | 4.7280 (0.0005) | 4.7273 (0.0002) | 4.7265 (0.0009) | 4.7273 (0.0010) | 4.7265 (0.0009) |
| c (Å) | 3.1796 (0.0006) | 3.1800 (0.0006) | 3.1797 (0.0002) | 3.1791 (0.0008) | 3.1796 (0.0008) | 3.1730 (0.0007) |
| Volume (Å) | 71.07 (0.02) | 71.08 (0.02) | 71.06 (0.01) | 71.02 (0.03) | 71.05 (0.03) | 71.02 (0.02) |
| $\chi^2$ (%) | 1.33 | 2.13 | 1.94 | 2.17 | 2.29 | 2.33 |
| $R_{Brag}$ (%) | 3.09 | 2.89 | 2.92 | 3.16 | 6.58 | 3.89 |
| $R_f$ (%) | 4.06 | 3.74 | 3.62 | 3.54 | 6.66 | 4.98 |
| $R_p$ (%) | 15.60 | 13.20 | 13.60 | 14.30 | 15.20 | 14.10 |
| $R_{\omega p}$ (%) | 17.50 | 15.00 | 15.80 | 16.20 | 18.00 | 16.50 |
| $R_{exp}$ (%) | 15.20 | 10.29 | 11.39 | 11.02 | 11.90 | 10.79 |
| Crystallite Size (nm) | 79.0 | 54.9 | 53.1 | 52.0 | 56.7 | 53.4 |

Here, the Scherrer constant k has a value dependent on the grain's form; however, we have assumed that the grains are circular, and thus, we have set the value of *k* to be 0.89; $\lambda$ is the wavelength of $CuK_\alpha$ radiation; $\beta$ is the line broadening; and $\theta$ is the Bragg's angle of X-ray diffraction, respectively. The value of the $S_c$ was evaluated by computing the values that correspond to a number of highly significant peaks, such as (110), (101), (200), and (211). The value of $S_c$ for undoped $SnO_2$ was estimated to be 79 nm, whereas the $S_c$ for samples of 2, 4, 6, 8, and 10 at.% Mg-doped $SnO_2$ was determined to be 55, 53, 52, 57, and 53 nm, respectively.

In order to understand the morphology of the prepared samples, one SEM image was obtained for the $Sn_{0.94}Mg_{0.06}O_2$ compound, and it is presented as Figure 4. The morphology obtained from the SEM micrograph reveals that the morphology of the sample is homogeneous with nearly spherical-shape nano-particles.

### 3.2. Raman Spectroscopy

Raman spectroscopy is a powerful non-destructive technique used to study various aspects of materials, including crystalline quality, structure, disorder, and defects in doped semiconductor oxides. Raman spectroscopy is based on the inelastic scattering of light, known as the Raman effect. The following primary vibration modes will often be produced by the $SnO_2$ lattice (without any flaws), according to numerous studies [56]:

$$\Gamma = A_{1g} + A_{2g} + B_{1g} + B_{2g} + E_g + A_{2u} + 2B_{1u} + 3E_u \quad (2)$$

Here, the Raman active modes are $A_{1g}$, $B_{1g}$, $B_{2g}$, and $E_g$, whereas the infrared active modes are $A_{2u}$ and $E_u$, and the inactive modes are $A_{2g}$ and $B_{1u}$. The simultaneous stretching of each Sn–O bond along a direction that maintains the crystal's symmetry is a typical aspect of the $A_{1g}$ mode. The $B_{2g}$ mode, on the other hand, includes asymmetric stretching of the Sn–O bond, i.e., stretching along a path that destroys the crystal's symmetry. Figure 5 displays the room temperature (30 °C) Raman spectra of Mg-doped $SnO_2$ compounds, with observations made in the 400 to 900 cm$^{-1}$ spectral range. The two detectable Raman active

modes of pure $SnO_2$ bulk materials at around 631 cm$^{-1}$ can be assigned to the $A_{1g}$ mode, and 769 cm$^{-1}$ can be assigned to the $B_{2g}$ mode. These measured Raman modes of $A_{1g}$ (symmetric Sn–O stretching) and $B_{2g}$ (asymmetric Sn–O stretching) can be indexed to the tetragonal phase of $SnO_2$ [53,59,60]. Table 2 provides the $A_{1g}$ and $B_{2g}$ mode's Raman peaks and intensities. As illustrated in Figure 6, the Lorentzian function fitting for these Raman active mode peaks ($A_{1g}$, $B_{2g}$) revealed that the resonance peaks of $A_{1g}$ and $B_{2g}$ are shifted towards the larger wave number (i.e., blue-shift) by increasing Mg doping concentrations. Here, the black lines represent the combined fitting for both peaks, whereas the green lines indicate the individual peak fitting. Moreover, it was noticed that the magnitude of $A_{1g}$ and $B_{2g}$ peaks' intensity changes in an irregular way by increasing the doping concentration of Mg in $SnO_2$. The change in the intensity might be related to the crystallite size of the synthesized materials, as observed in other studies. A study by J. Zuo et al. [60] reported that intensity peaks have a dependency on the crystallite size. Additionally, it was noticed that no apparent peak corresponding to the $E_g$ mode of vibration, usually located at 475 cm$^{-1}$ is observed. Thus, the development of the tetragonal rutile phase of $SnO_2$ was confirmed by both Raman spectra and XRD patterns.

**Table 2.** Raman shift and height of the primary active Raman vibration modes ($A_{1g}$ and $B_{2g}$) of the rutile phase of Mg-doped $SnO_2$ compound. The center ($A_{1g}$) represents the peak position of $A_{1g}$ mode, whereas the center ($B_{2g}$) represents the peak position of $B_{2g}$. The intensity ($A_{1g}$) represents the intensity observed for the $A_{1g}$ peak, whereas the intensity ($B_{2g}$) represents the intensity observed for the $B_{2g}$ peak of every Mg-doped $SnO_2$ sample. FWHM ($A_{1g}$) represents the full width at half maxima of $A_{1g}$ peaks.

| Sample/Parameters | 0% | 2% | 4% | 6% | 8% | 10% |
|---|---|---|---|---|---|---|
| Center ($A_{1g}$) (cm$^{-1}$) | 630.6 | 632.2 | 632.9 | 633.3 | 634.0 | 633.7 |
| Intensity ($A_{1g}$) | 12,521.0 | 18,847.1 | 11,630.5 | 23,924.9 | 19,297.4 | 36,030.4 |
| Center ($B_{2g}$) (cm$^{-1}$) | 769.3 | 773.7 | 774.15 | 772.6 | 771.8 | 774.0 |
| Intensity ($B_{2g}$) | 1798.6 | 4233.4 | 2342.7 | 4715.0 | 3288.8 | 8060.3 |
| FWHM ($A_{1g}$) (cm$^{-1}$) | 11.9 | 10.5 | 10.9 | 10.0 | 9.9 | 10.9 |

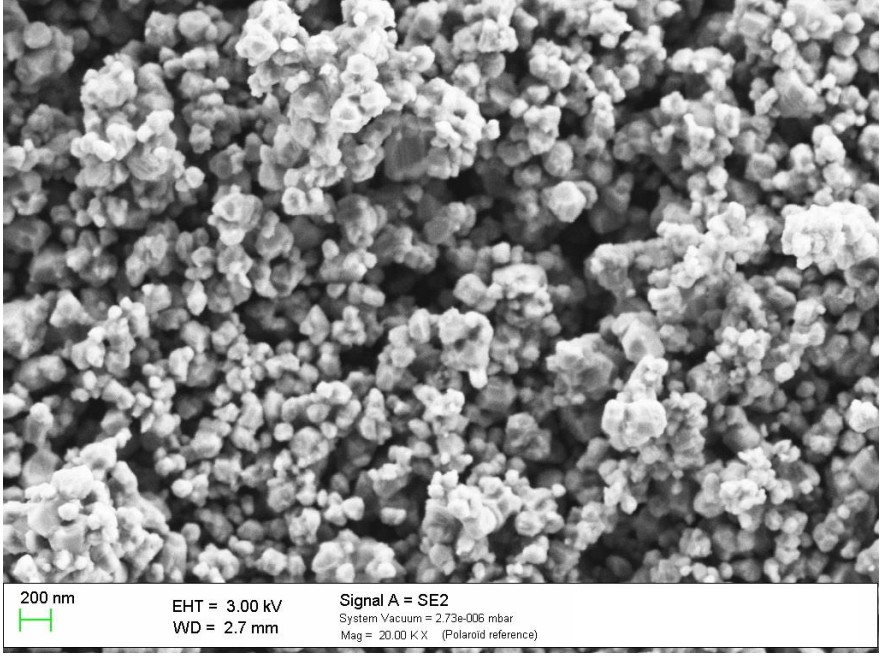

**Figure 4.** The scanning electron microscopy image for $Sn_{0.94}Mg_{0.06}O_2$ compound.

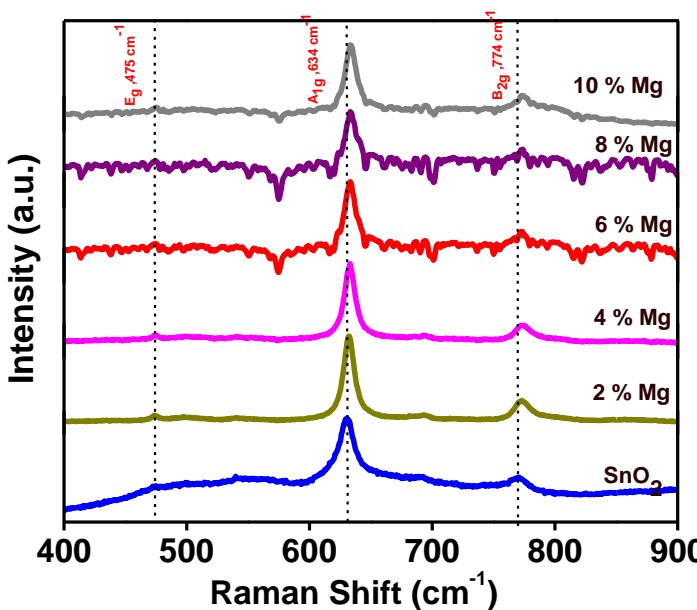

**Figure 5.** Raman spectra of SnO$_2$ compounds doped with varying concentrations of Mg.

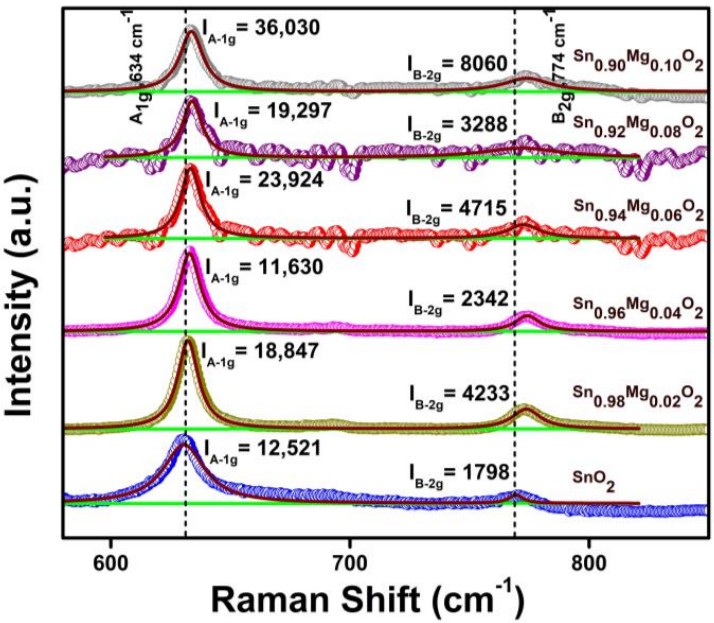

**Figure 6.** Lorentzian fitting of the Mg-doped SnO$_2$ Raman spectra. The black lines represent the combined fitting for both peaks, whereas the green lines indicate the individual peak fitting. I$_{A-1g}$ represents the intensity observed for the A$_{1g}$ peak, whereas I$_{B-2g}$ represents the intensity observed for the B$_{2g}$ peak of every Mg-doped SnO$_2$ sample.

### 3.3. Optical Property

The optical absorption spectra of the Sn$_{1-x}$Mg$_x$O$_2$ compounds with varying compositions (x = 0, 0.02, 0.04, 0.06, 0.08, and 0.10) were measured using a UV-Vis spectrophotometer. Figure 7 shows the measurement of absorbance spectra of Mg-doped SnO$_2$ compounds with 0, 2, 4, 6, 8, and 10 at.% of Mg. The experiments were conducted in the wavelength range of 200–800 nm. All samples have a UV absorption edge at 250 nm in their absorption spectra. This may be due to optical transitions between the valence and conduction bands (O$_{2p}$ to Sn$_{3d}$), which causes the tin oxide's intrinsic band gap to absorb light [61]. However, spectral transmittance data between 360 and 800 nm clearly demonstrate transparency.

The $SnO_2$ UV absorption edge in these compounds is located at 250 nm, making them effective at blocking light with a wavelength of less than 360 nm. With increased Mg content, the transmittance value rises from 83% (for pure $SnO_2$) to 91% (for $Sn_{0.90}Mg_{0.10}O_2$). The enhancement in transmittance with increasing doping concentrations may result from optical scattering due to uniformly distributed particles [62]. The energy band gaps of the materials were determined by applying the following formula:

$$(\alpha h\nu)^2 = (h\nu - E_g) \tag{3}$$

which was developed by Tauc, Mott, and Davis, usually called Tauc's formula, applicable to direct band gap semiconductors. The absorption coefficient is denoted by $\alpha$, Planck's constant is denoted by $h$, the photon's frequency is represented by $\nu$, and the energy band gap of the semiconductor is written as $E_g$ [62]. A Tauc's plot was generated from the absorption spectra by plotting $(ah\nu)^2$ versus photon energy $h\nu$. A linear section of Tauc's plot in the higher energy region was selected, and linear fitting was performed. Finally, the intercept at the $h\nu$ axis was taken as the band gap ($E_g$). Figure 8 shows Tauc's plots for all the samples. Figure 8a shows the band gap energy of undoped $SnO_2$ is 3.69 eV, comparable to the typical value of pure $SnO_2$ (3.6 eV). The energy band gaps for $SnO_2$ compounds with 2, 4, 6, 8 and 10 at.% of Mg doping, respectively, are shown in Figure 8b–f. The associated band gap values are 3.72 eV, 3.72 eV, 3.71 eV, 3.71 eV, and 3.73 eV, respectively. The variations in the band gap along with the error of all Mg-doped $SnO_2$ compounds are shown in Figure 8g. Here, the presented errors are taken from the linear fitting of the curve. Therefore, the predicted band gap of Mg-doped $SnO_2$ compounds exhibits a band gap widening (blue shift) up to 2% of Mg doping, followed by a decrease for higher doping levels. The Moss–Burstein effect, often referred to as the Burstein–Moss shift, can explain the expansion of the band gap caused by up to 2% of Mg doping. This theory states that when the number of charge carriers increases, the Fermi level in the conduction band is completely filled at a given dopant concentration. The extra excited electrons are supposed to move up into the conduction band, which is above the Fermi level. This scenario causes the material's band gap to expand [62]. On the other hand, the observed slight decrease in band gap energy (red shift) from 2% to 8% might be due to the substitution of $Mg^{2+}$ ions in the $SnO_2$ host material. Further, the energy band gap of Mg-doped $SnO_2$ is widened from 8% to 10% doping of Mg. Similar results were seen in epitaxial Mg-doped $SnO_2$ thin films prepared via radio-frequency magnetron sputtering and in other work [22,51,55,56]. Wu et al. reported that the optical band gap of these materials was found to increase with Mg doping. The observed ferromagnetism was claimed to be induced by the holes created by Mg on the substitutional site [22]. He et al. [55] studied the effect of Mg doping on the structural, optical, and electrical properties of $SnO_2$ thin films deposited via the electron beam technique. It was observed that these films exhibit the optical transmittance of 83% in the visible region. The band gap of the films was found to change from 3.49 to 3.78 eV [55]. Ali et al. [56] studied the effect of Mg doping on the structural and optical properties of aerosol-assisted chemical-vapor-deposited $SnO_2$ thin films. It was noticed that the optical transmission increased from 54 to 78%. The optical band gap was found to decrease with an initial doping concentration, followed by an increase upon further increasing the Mg doping concentration. Narzary et al. reported that $Sn_{0.94-y}Ag_{0.06}Mg_yO_2$ (with y = 0, 0.03, 0.06, 0.09, 0.12) exhibits RTFM, with coercivity ($H_C$) values lying within the range 10–50 Oe [27]. These materials were found to show excellent optical transparency [27].

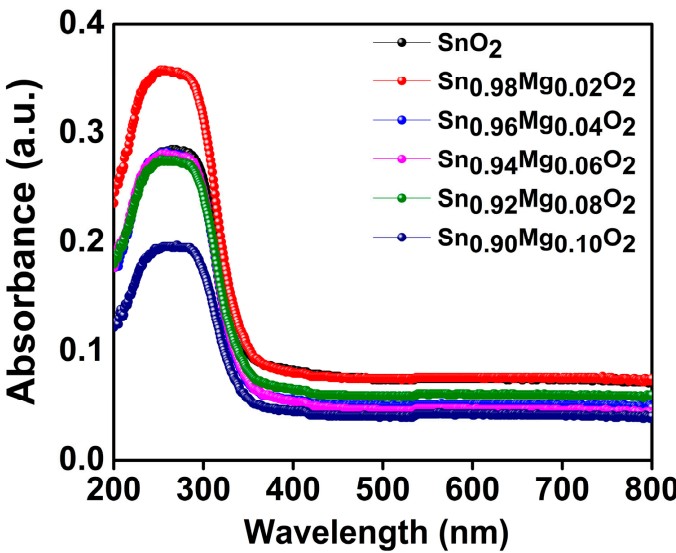

**Figure 7.** UV-Visible absorption spectra of $Sn_{1-x}Mg_xO_2$ with x = 0, 0.02, 0.04, 0.06, 0.08, and 0.10 compounds.

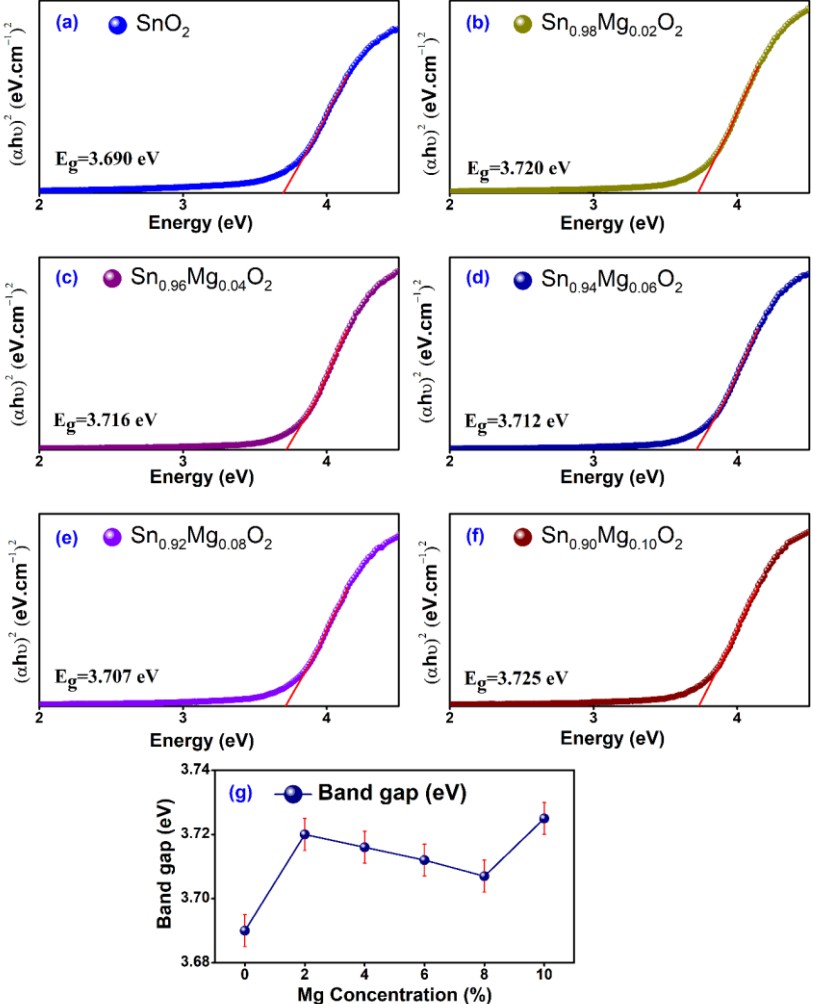

**Figure 8.** Tauc's plots for $Sn_{1-x}Mg_xO_2$ compounds (x = 0, 0.02, 0.04, 0.06, 0.08, and 0.10) are shown in (**a–f**). Band gap shifts (along with error bar) in $SnO_2$ due to Mg doping are shown for a range of Mg concentrations in (**g**). The errors presented here are taken from the linear fitting of the curve.

## 4. Conclusions

In conclusion, we conducted an exhaustive study of the crystal structure, Raman, and optical properties of Mg-doped $SnO_2$ samples synthesized via the solid-state reaction technique. According to the results of the XRD analysis, the samples were crystallized into a tetragonal-rutile-phase structure of the $SnO_2$ compound. The refined XRD patterns show that the amount of magnesium that is incorporated into tin oxide results in a decrease in the value of the lattice parameters. The crystalline size of the synthesized samples was measured using the Debye–Scherer formula, and the results showed that it ranges from 52 to 79 nanometers. The subsequent analysis with a Raman spectra also indicates that the tetragonal rutile crystal structure is present in the material. The band gap of the produced compounds broadens up to a doping level of 2% Mg, and then it begins to narrow again as the doping level increases up to 8%. Further, it widens from 8% to 10% doping of Mg due to the substitution of Mg in the interstitial site of tin oxide. In addition, based on the optical transmittance spectra, it is noticed that adding Mg into $SnO_2$ raises the transmittance value from 83% to 91%, which is an improvement in optoelectronics devices. In areas of optoelectronics where excellent optical transparency is required, the produced polycrystalline $Mg:SnO_2$ compounds can be employed as constituent materials.

**Author Contributions:** K.K.S.: Investigation, Methodology, Formal Analysis, Writing Manuscript; P.P.S.: Formal Analysis, Review and Editing; R.N.: Investigation, Formal Analysis; A.M.: Investigation; M.G.: Resources, Review and Editing; V.G.S.: Resources, Review and Editing; D.K.: Resources, Review and Editing; S.K.S.: Conceptualization, Methodology, Visualization, Writing—Review and Editing, Supervision. All authors have read and agreed to the published version of the manuscript.

**Funding:** S.K.S. acknowledges the financial support from UGC-DAE CSR through a Collaborative Research Scheme. This research was funded by UGC-DAE CSR grant number CRS/2021-22/01/364. The APC was funded by professional development allowance given by CIT Kokrajhar.

**Data Availability Statement:** The data that support the findings of this study are available within the article.

**Acknowledgments:** K.K.S. would like to express special thanks to V.G.S. (Centre-Director) and Ajay K. Rathore (Scientific Officer-F) for helping with the Raman spectroscopy at UGC-DAE-CSR Indore. Further, K.K.S. would like to express special thanks to M.G. (Scientist-G), Layanta Behera (Scientific Assistant-E) for helping with the XRD measurements at UGC-DAE-CSR Indore.

**Conflicts of Interest:** The authors declare no conflict of interest.

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
