# Peer review of "Crystal Structure, Raman Spectroscopy and Optical Property Study of Mg-Doped SnO2 Compounds for Optoelectronic Devices"

_crystals, doi:10.3390/cryst13060932_

Round 1
Reviewer 1 Report
The authors evaluated the characteristics of Mg-doped SnO2 compounds for optoelectronic devices, which is believed to be beneficial for applied research on Mg-doped SnO2. however, I cannot recommend its publication in its current form as it has critical errors. I have noted my specific concerns below.
1. Why is the range limited to 0.1%? What is the doping limit range, and is there a possibility of further enhancing the characteristics with higher doping conditions?
2. Was the baseline flatly corrected in the XRD analysis? If so, please describe the method. In addition, the Rietveld refinement should include data from the high-angle region where the baseline slope has relatively minimal influence. The graph should provide counts instead of arbitrary units for intensity on the vertical axis.
3. I think that in either case Interstitial and substitutional site, if the ionic radius increases, the lattice size should also increase. However, the authors explain a decrease in lattice size in line 147 of the main text. To ensure readers' understanding, it is necessary to provide a clear and comprehensible explanation along with a figure.
4. If abbreviations have been defined, they should be consistently used throughout the text.(For example, in line 153, an abbreviation was defined, and then it was used in line 158. However, in line 160, the abbreviation is defined again.)
5. What does the error bar value in Figure 3 represent? Is it standard deviation or deviation?
6. The caption of Table 1 needs to be corrected rightly.
7. What is the difference between "ambient temperature" and "room temperature"? Please provide an exact expression in degrees Celsius.
8. In the text, various characterization results are listed, and it is necessary to specify the exact type of intensity. Please provide clarification including “intensity (Raman?)” in line 195.
9. The authors claim, based on reference 60, that the Raman intensity increases when the lattice size decreases. However, pure SnO2 exhibits the largest lattice size, but the Raman intensity of A1g is smallest in the sample doped with 4%. A clear explanation is required regarding this issue.
10. Please provide additional clarification regarding which Raman results correspond to interstitial and substitutional sites in lines 200-201.
11. The actual composition of the prepared material should be investigated.
12. The authors mentioned the SEM and XPS analyses in the Acknowledgments, and it is necessary to provide the corresponding data for these results.
13. The authors cite reference 58 to explain both the increase and decrease of the bandgap after doping. However, the referenced study only explains the decrease of the bandgap after doping.
Reviewer 2 Report
I found this article interesting enough for publication. Nevertheless, some clarification is important to consider.
Some questions appear to be related to the two essential results.
The authors made a detailed study of the compounds by XRD. Crystallographic parameters have been determined. Why is sample 6 very different from others, as seen in Fig. 3? It should be commented on in more detail.
The authors suggest an intersite localization of the Mg ions in the lattice. It leads to the creation of defects and, as a result, to a shift in the cutoff of the fundamental absorption. The authors obtained the opposite result, as seen in Fig. 7.
XDR data can be used to define Ronthen density and see a correspondence to the Vegard law.
The procedure for the optical transmittance measurements with powdered samples has to be described in more detail. The scale of the ordinate axis in Fig. 6 (a) should be digitized. In the absence of the scale numbers, it is difficult to say about the correct application of equation 3.
A comment on the possible correlation of the data in Fig. 3 and Fig. 7(h) should be provided.
A major revision is required.
Reviewer 3 Report
The paper titled “Crystal Structure, Raman Spectroscopy and Optical Property 2 Study of Mg-doped SnO2 Compounds for Optoelectronic De-3 vices”: by K. K. Singha et al. requires significant revisions prior to its publication in the journal "Crystals." Overall, it is a nice demonstrate of the data provided, but I have a few questions/comments.
1. One of the primary inquiries pertains to the resolution of your XRD experiments. It is worth verifying the accuracy and precision of your measurement and analysis, particularly in relation to significant figures, as the authors presented values with an extensive number of decimal places, extending to the hundred thousandth angstrom.
2. Regarding the characterization of materials in the experimental section, it would be advisable to separate it into XRD analysis and optical characterization for clarity. In the Raman study, there are certain missing details, such as information about the acquisition time and the method employed for baseline subtraction. Additionally, it would be beneficial to include a discussion on the spectral resolution of the Raman measurements.
3. To enhance the characterization process, providing morphology and elemental analysis would be beneficial for readers to gain a better understanding of this research.
4. Overall, the captions accompanying the figures have insufficient legibility. In Figure 2 and 4, for instance, the letters are too small to discern. I suggest reviewing all figures and their respective captions, ensuring that the text is appropriately sized to improve visual clarity.
5. Regarding Table 2, it would be helpful to clarify the meaning of the term "height." Assuming it represents intensity or area, it is important to specify the unit of measurement, as it should not be in wavenumbers.
6. The authors state that the peak of the Eg mode does not exhibit a noticeable shift in their Raman spectra. However, I am unable to observe any significant peaks around 475 wavenumbers. It is crucial for them to discuss which peaks are present, shifted, or display changes in intensity.
7. Figure 5 lacks an explanation for the red and green lines. It is recommended to either include detailed figure captions or provide appropriate explanations within the manuscript. Additionally, please ensure consistent usage of the terms "spectrum" and "spectra" throughout the entire manuscript, as they represent singular and plural forms, respectively.
8. In Figure 6, both the absorption and transmission spectra are duplicated. It may be unnecessary to include both. For the absorption spectra, it would be beneficial to add intensity values on the y-axis. The authors claimed that the bandgap undergoes a blue shift with increasing Mg doping concentration. Further explanation is required regarding the resolution of the measurement, their analysis methodology, and the physical significance of bandgap engineering. Additionally, the calculation method for the error bars in Figure 7-g needs to be explained. Furthermore, the caption of Figure 7 should be revised to include (h), as it is currently missing.
Round 2
Reviewer 1 Report
The authors have addressed most of the points well. However, some points have not been addressed yet. If they resolve the following points, I think it would be ready for publication.
1. For reliability improvement in Rietveld analysis, it is generally recommended to have intensities of at least 10,000 counts. Additionally, it is crucial to include information regarding high-angle regions above 80 degrees. However, the authors conducted their analysis within a limited range of 20-80 degrees with counts ranging from 4,000 to 5,000. An explanation is required for the discrepancy between the generally used condition and the approach taken by the authors.
Author Response
Question 1: For reliability improvement in Rietveld analysis, it is generally recommended to have intensities of at least 10,000 counts. Additionally, it is crucial to include information regarding high-angle regions above 80 degrees. However, the authors conducted their analysis within a limited range of 20-80 degrees with counts ranging from 4,000 to 5,000. An explanation is required for the discrepancy between the generally used condition and the approach taken by the authors.
Answer: Dear Reviewer! Thank you for your valuable remark about the reliability of the Rietveld analysis of XRD with low counts of intensities. However, we would like to emphasize that it is clearly evident from the analyses of the XRD refinement (Table 1) that the fitment factor ê“2 (%) of refinement ranges only between 1.33-2.33, which is an indication of the very good fitting.
Moreover, in most of the literature on Oxide materials, you will see that the XRD measurements are performed up to 80 degrees only. Thus, the chosen range of 20-80 degrees for XRD analyses is sufficient to study the crystallographic information of the presented materials. Thank you for your understanding.
Reviewer 2 Report
It can be published in a present form.
Author Response
Thank you so much for accepting it.
Reviewer 3 Report
Please provide the data analysis regarding the morphology of the Mg-doped SnO2 compounds. Specifically, I am interested in knowing whether the analysis indicates the presence of micro- or nano-particles, or if the morphology exhibits film-like bulk properties. Your insights into the morphology of the Mg-doped SnO2 compounds would greatly contribute to a better understanding of the material's structural characteristics.
The authors have claimed their XRD measurement and data analysis, stating that they can explain with accuracy to the hundred thousandths angstrom. It would be valuable to compare these results with previously reported lattice constants and provide an explanation as to how their measurement achieves ten times better accuracy. Conducting a systematic error analysis would be an effective approach to substantiate their claim. By thoroughly examining and quantifying systematic errors, the authors can provide evidence supporting their improved accuracy.
I would like to bring to your attention a concern regarding the Materials and Methods section. Specifically, in line 129 of the manuscript, the authors referred to the "spectral resolution" of Raman spectroscopy. However, it appears that there may be an error, and the intended term should be "spatial resolution" instead of "spectral resolution." It would be greatly appreciated if you could provide the accurate spectral resolution values for the Raman measurements discussed in the manuscript. Please ensure that the discussion of these values includes the appropriate significant figures. Additionally, the authors state the Raman spectra without any baseline correction, in line 130. However, in Figure 4, the Raman spectra seem to be baseline corrected. If not, please explain the negative shapes of baselines in Mg-doped SnO2. In Table 2, please define the meaning of intensity, and add the baseline and intensity values in Figures 4 and 5. Please provide the fitting result with fitting quality-related values in the supplementary section.
I have a question regarding Figure 6. I am uncertain about the rationale behind presenting the UV-vis spectra using two different modes: absorption and transmission. Typically, absorption is represented as the logarithmic function of transmission. Please explain the reason for including both modes in the figure and elaborate on the specific data analysis that the authors intended to conduct using these representations. Clarifying these points would greatly enhance my understanding of the figure and its purpose.
I have a concern regarding Figure 7. In the figure, the authors mention "the error from the linear fitting of the curve," but it is unclear what exactly is meant by the term "fitting error." Could you please provide further clarification on whether the authors are referring to the value obtained from the 95% prediction interval, or any other specific regression error related to the x-intercept? Additionally, it would be helpful if you could explain in detail how the experimental data were fitted and how the fitting errors are being explained. Elaborating on these aspects would greatly assist in understanding the analysis conducted and the interpretation of the fitting errors.
Author Response
Thank you very much for your further valuable feedback on our manuscript.
Question 1: Please provide the data analysis regarding the morphology of the Mg-doped SnO2 compounds. Specifically, I am interested in knowing whether the analysis indicates the presence of micro- or nano-particles, or if the morphology exhibits film-like bulk properties. Your insights into the morphology of the Mg-doped SnO2 compounds would greatly contribute to a better understanding of the material's structural characteristics.
Answer: In the revised manuscript, we have added one SEM image (as Fig. 4) to understand the morphology of the Mg-doped SnO2 compounds. The image indicates the presence of the nano-particles and it is discussed in the line no. 193-196.
Question 2: The authors have claimed their XRD measurement and data analysis, stating that they can explain with accuracy to the hundred-thousandths angstrom. It would be valuable to compare these results with previously reported lattice constants and provide an explanation as to how their measurement achieves ten times better accuracy. Conducting a systematic error analysis would be an effective approach to substantiate their claim. By thoroughly examining and quantifying systematic errors, the authors can provide evidence supporting their improved accuracy.
Answer: We are thankful to you for pointing out this error in the manuscript, which went unnoticed from our end. While writing the error analyses of lattice parameters in Table-1, by mistake we have written the error with one extra zero, which resulted in great confusion.
Now, in this re-revised manuscript, we have taken care of the significant figures and uncertainties in the derived quantities. We have followed the accepted rules for presenting the data i.e. the data should only be reported in terms of as many significant figures as are consistent with the estimated error and only one uncertain digit is to be reported for a measurement.
Question 3: I would like to bring to your attention a concern regarding the Materials and Methods section. Specifically, in line 129 of the manuscript, the authors referred to the "spectral resolution" of Raman spectroscopy. However, it appears that there may be an error, and the intended term should be "spatial resolution" instead of "spectral resolution." It would be greatly appreciated if you could provide the accurate spectral resolution values for the Raman measurements discussed in the manuscript. Please ensure that the discussion of these values includes the appropriate significant figures. Additionally, the authors state the Raman spectra without any baseline correction, in line 130. However, in Figure 4, the Raman spectra seem to be baseline corrected. If not, please explain the negative shapes of baselines in Mg-doped SnO2. In Table 2, please define the meaning of intensity, and add the baseline and intensity values in Figures 4 and 5. Please provide the fitting result with fitting quality-related values in the supplementary section.
Answer:
Thank you for your kind suggestions. We have corrected the term "spatial resolution" instead of "spectral resolution." in line 130 of the manuscript. Moreover, we have added the spectral resolution values with the appropriate significant figure for the Raman measurements in the revised manuscript. Again, we would like to mention that the spectrum was collected without any baseline correction. However, we are unable to comment on the negative shapes of baselines in Figure 4 (the Raman spectra). In the revised manuscript, we have defined the meaning of intensity in Table 2 and added the intensity values in Figure 6. Moreover, we have added the fitting result with fitting quality-related values in Table 2 itself.
Question 4: I have a question regarding Figure 6. I am uncertain about the rationale behind presenting the UV-vis spectra using two different modes: absorption and transmission. Typically, absorption is represented as the logarithmic function of transmission. Please explain the reason for including both modes in the figure and elaborate on the specific data analysis that the authors intended to conduct using these representations. Clarifying these points would greatly enhance my understanding of the figure and its purpose.
Answer: To avoid confusion to the reader, we have decided to remove the transmission spectrum.
Question 5: I have a concern regarding Figure 7. In the figure, the authors mention "the error from the linear fitting of the curve," but it is unclear what exactly is meant by the term "fitting error." Could you please provide further clarification on whether the authors are referring to the value obtained from the 95% prediction interval, or any other specific regression error related to the x-intercept? Additionally, it would be helpful if you could explain in detail how the experimental data were fitted and how the fitting errors are being explained. Elaborating on these aspects would greatly assist in understanding the analysis conducted and the interpretation of the fitting errors.
Answer:
Now, these details have been provided in lines 269-272 and 277-279 of the re-revised manuscript.